# Effects of Rearing Aviary Style and Genetic Strain on the Locomotion and Musculoskeletal Characteristics of Layer Pullets

**DOI:** 10.3390/ani11030634

**Published:** 2021-02-27

**Authors:** Amanda Pufall, Alexandra Harlander-Matauschek, Michelle Hunniford, Tina M. Widowski

**Affiliations:** 1Department of Animal Biosciences, Ontario Agricultural College, University of Guelph, Guelph, ON N1G 2W1, Canada; apufall@uoguelph.ca (A.P.); aharland@uoguelph.ca (A.H.-M.); 2Burnbrae Farms, Lyn, ON K0E 1M0, Canada; mhunniford@burnbraefarms.com

**Keywords:** aviary, rearing, poultry welfare, locomotion, musculoskeletal development

## Abstract

**Simple Summary:**

Consumer trends have led to the popularisation of eggs from hens housed in non-cage systems. Aviaries are common non-cage systems that consist of multi-tiered structures with essential resources located within. Being a ground-dwelling species, laying hens must learn when they are young to navigate aviaries, and develop the strength to do so. However, there are different styles of aviaries with varying degrees of structural complexity in which young laying hens (pullets) are reared. This study examined the differences in locomotion and musculoskeletal characteristics of white- and brown-feathered strains of pullets housed in three different styles of rearing aviaries (low, moderate, and high structural complexity) on commercial farms. Pullets in the most complex system spent the most time locomoting, and white-feathered strains in the highly complex systems performed the most jumps and flights. While there were no differences in muscle weights between aviary styles, white-feathered strains, and pullets in the most complex aviary had stronger leg bones than brown-feathered strains and pullets in the least complex system. These results indicate that the style of the aviary in which pullets are reared can affect their bone strength, most likely through differences in the load-bearing exercises performed.

**Abstract:**

Previous research indicates that the musculoskeletal development of pullets is improved when pullets are reared in aviaries compared to conventional rearing cages. However, there are considerable differences in rearing aviary design. To measure locomotion and musculoskeletal development of brown (*n* = 7) and white-feathered (*n* = 8) strains of pullets, 15 commercial flocks in three styles of rearing aviaries differing in structural complexity (*n* = 5 per style) were visited three times: 25.9 ± 6.67, 68.0 ± 4.78, and 112.1 ± 3.34 days of age. Locomotion (duration of standing, sitting, walking, running, flying, and rates jumping, flying, group running and walking) was analysed from videos recorded three times per day: at the beginning, middle, and end of the light cycle. Pullets for dissection were taken on visits 2 and 3. Pullets in the most complex system (style 3; S3) spent the most time locomoting throughout rearing (*p* < 0.05). Pullets in S3, particularly white-feathered strains, performed the highest rate of vertical transitions (*p* < 0.05). There were no differences in any of the proportional muscle weights between aviaries styles (*p* > 0.05) despite the differences in locomotion. White-feathered strains, however, had proportionally heavier pectoralis major (*p* < 0.0001), pectoralis minor (*p* < 0.0001), and lighter leg muscles (*p* < 0.0001) than brown-feathered strains. White-feathered strains and pullets in S3 also had proportionally stronger tibiae and femurs than brown-feathered strains and pullets housed in the least structurally complex system (style 1; S1) (*p* < 0.05). However, there were no differences found in the breaking strength of the radius and humerus between strain colours or aviary styles (*p* < 0.05). Therefore, strain, as well as differences in rearing aviary design, can affect the types of locomotion that growing pullets perform, which may, in turn, impact their skeletal development.

## 1. Introduction

In recent years, alternative housing systems, including enriched colony cages and non-cage systems such as aviaries, have started to become more commonplace in the egg-laying industry. In 2016, the Egg Farmers of Canada announced an industry-wide transition to alternative systems by 2036 [1]. Multiple large retailers have also made commitments to purchase eggs from hens housed in non-cage systems [2]. Non-cage systems improve some aspects of hen welfare as they provide hens with more opportunities to perform highly motivated behaviour patterns such as dust bathing [3], perching [4], and laying their eggs in nests [5]. However, for hens to be successful in adult aviaries, they must be reared in an aviary that is similarly as complex as their adult housing system [6].

Rearing pullets in a complex environment with multiple vertical levels has been shown to improve pullets’ spatial cognition [7], working memory [8], and musculoskeletal characteristics [9,10,11] compared to rearing pullets in conventional cages. Aviary-reared pullets have also been found to have proportionally heavier breast muscles, larger keels, and stronger long bones of the legs and wings than conventionally reared pullets [9]. Even the addition of a perch to a conventional cage has been shown to increase the pullets’ leg muscle weight and tibia bone mineral content at the end of rearing [12]. The differences in the musculoskeletal characteristics between pullets reared in the different housing systems were likely due to differences in locomotion during development.

Applying load to the bone either through weight-bearing loads or muscle contractions that strain the bone will induce bone remodelling, thus increasing bone strength [13,14]. However, bone remodelling is specific to the site where the load is applied [15,16], and not all types of exercise contribute sufficient load to induce bone remodelling [17]. Few specific locomotory patterns in laying hens have been studied regarding their effects on musculoskeletal development. Judex and Zernicke [18] found that jumping increased the bone formation of the tarsometatarsus of immature roosters. While not directly tested, Casey-Trott et al. [9] hypothesised that the increased keel size of aviary-reared pullets was due to increased wing-assisted locomotion. Wing use involves the pectoral muscles, which are attached to the keel [19]; therefore, increased wing use may increase pectoral muscle contraction against the keel, which could, in turn, induce keel bone formation.

There are also inherent differences, both in regard to musculoskeletal characteristics and locomotory behaviours, between strains of laying hens, specifically between white- and brown-feathered strains. Fawcett et al. [20] found that white-feathered pullets reared in enriched cages had proportionally stronger femurs and tibiae than brown-feathered pullets reared in the same environment. Furthermore, the authors reported that white-feathered pullets had proportionally heavier pectoral muscles and lighter leg muscles than brown-feathered pullets. As there was no observed difference in locomotion between the pullets [21], the differences in musculoskeletal characteristics were likely due to differences in body conformation between strains. Furthermore, when reared in pens outfitted with perches and platforms, Kozak et al. [22] reported that white-feathered strains performed more aerial locomotion than brown-feathered strains.

While it appears as though early life environment can affect the locomotion of pullets, to date, only one study has examined how rearing environment impacts locomotion. Colson et al. [23] found that, during rearing, pullets in aviaries with feeders on the floor performed more numerous and longer aerial transitions than pullets in aviaries with feeders within the system. After transfer to adult housing, floor-reared hens performed fewer long aerial transitions than both groups of aviary-reared pullets. However, to date, there are no studies comparing different commercially available rearing aviaries, as there are multiple styles on the market. In the presently studied rearing aviaries, chicks start in a brooding compartment with varying amounts of structural complexity. After a few weeks, producers open the compartments, providing pullets access to a litter floor, as well as additional vertical tiers, perches, terraces, and ramps [24].

This study aims to investigate how rearing aviary style and strain (white vs. brown) affects the locomotion behaviours and musculoskeletal characteristics of pullets throughout the rearing period. We hypothesised that white-feathered strains and pullets reared in the most structurally complex system would perform more locomotion, have proportionally heavier pectoral muscles, lighter leg muscles, and stronger long bones compared to brown-feathered strains and those experiencing a greater degree of confinement during the brooding period.

## 2. Materials and Methods

### 2.1. Animals

A total of 15 pullet flocks from 11 different commercial farms in Alberta and Ontario were visited from May 2018 to March 2019. Fourteen flocks were on commercial farms with flock sizes ranging from 6000 to 43,000 birds. One additional flock of 700 birds was located at the University of Guelph Arkell Poultry Research Station. The average commercial flock size was approximately 21,150 birds. Eight flocks were white-feathered strains, and seven were brown-feathered strains. Flocks were vaccinated, fed, and provided light according to individual farm management. Light intensity varied across flocks and was not measured.

There was a wide range of stocking densities across flocks, both during the brooding period (floor and platform space within the compartment) and after brooding compartments were opened and pullets had access to additional tiers, terraces and litter. In S1, stocking densities ranged from 195.68 cm^2^/bird to 199.24 cm^2^/bird to when brooding compartments were closed and from 487.21 cm^2^/bird to 709.21 cm^2^/bird when compartments were opened. In S2, stocking densities ranged from 128.40 cm^2^/bird to 183.43 cm^2^/bird when brooding compartments were closed and from 358.76 cm^2^/bird to 843 cm^2^/bird after the compartments were opened. In S3 values ranged from 306.92 cm^2^/bird to 410.94 cm^2^/bird when closed, and from 438.12 cm^2^/bird to 890.62 cm^2^/ bird when opened. All stocking densities are approximate, calculated from the number of chicks placed and usable floor space of system, terraces and litter areas, from measurements taken on farm.

### 2.2. Housing and General Husbandry

Pullets were housed in one of three general rearing aviary styles. Aviary styles were differentiated primarily by the brooding compartments in which chicks were placed. Details of models, strain of pullets, and additional management practices can be found in the Appendix A.

#### 2.2.1. Style One (S1)

Chicks were placed in single level wire-floored brooding compartments furnished with an internal feed trough, one line of nipple drinkers and two perches (Figure 1a). The system had three tiers, with chicks brooded in the bottom two tiers. Both perches spanned the length of the brooding compartment. The length of the brooding compartment varied from 148.5 cm to 231 cm. One perch was located above the feeder and another perch was mounted above the drinker and was raised as the pullets grew. Producers opened the compartments at 37.6 ± 8.85 days of age, providing access to all three tiers, litter floors, terraces, ramps, and additional perches over the litter and within the top third tier. Drinker lines were available on the bottom two tiers, and feed troughs were available on all three tiers. Ramp availability varied by flock and visit number.

#### 2.2.2. Style Two (S2)

Chicks were placed in wire-floored brooding compartments furnished with an internal feed trough, one line of nipple drinkers, three round metal perches and a raised platform at the rear of the compartment (Figure 1b). The system had two identical tiers, with chicks brooded in the top tier. The length of the brooding compartment was 240 cm. All three perches spanned the length of the compartment. One perch was a fixed height above the feeder (43 cm). A second perch was mounted above the drinker and was raised as the pullets grew. The third perch was a fixed height above the platform (61 cm from the system floor). The raised platform was at the rear of the compartment (36 cm off the system floor), spanning the entire length at a height of 36 cm, and had a width of 28 cm. Producers opened the compartments at 26.8 ± 2.68 days of age, providing access to two tiers of identical compartments, a litter floor, and an external perch over the litter between the two tiers. Pullets were not able to move between aisles. Ramp availability varied by flock and visit number.

#### 2.2.3. Style Three (S3)

Chicks were placed in an open-concept system that typically spanned the length of the barn (Figure 1c). The systems were enclosed with hinged vertical panels made of a wire mesh or plastic grid flooring material that could be rotated to open the system and become additional ramps or platforms. Each panel measured from 229 cm to 240 cm. A minimum of 6 perches spanned the length of the system (length of the system varied for each flock depending on length of the barn). Perches mounted above the drinkers were adjustable and were raised as the chicks grew. A centre platform (58 cm to 60 cm in width) spanned the length of the system and had a drinker line above it. The centre platform and drinker lines were raised as the chicks grew. Ramps availability varied by flock and visit number.

Producers opened the systems at 54.6 ± 20.50 days of age providing access to a litter floor, ramps, and additional platforms and terraces.

### 2.3. Data Collection: Behaviour

#### 2.3.1. Video Collection

Videos were collected during all three visits: before the brooding compartments were opened at 25.9 ± 6.67 days of age (visit one; V1); in the middle of rearing at 68.0 ± 4.78 days of age; (visit two; V2); near the end of the rearing period at 112.1 ± 3.34 days of age (visit three; V3). Three two-hour recordings were taken at each visit. The first recording started approximately 30 min after the lights turned on. The second recording was scheduled such that one hour into the recording was the middle of the light cycle. The final two-hour recording period finished 30 min before the lights began to dim.

Video cameras (Sony Handycam HDR-X240 and HDR-X110, Tokyo, Japan) outfitted with wide-angle conversion lenses were set up systematically in barns to capture views both within the system and over the litter floor (Figure 2). Cameras were set up to view the same brooding compartments or panel sections at each visit. Cameras were mounted on monopods (Digiant MP-3606 Professional Video Monopod 70”, Zhejiang, China) or attached to fixtures on the system with gorilla mounts (Lammcou Bendable Gorilla Portable Tripod, China). For the first visit (V1), cameras were set up to view behaviours within the brooding compartments in S1 and S2, and within the system in S3. During visits 2 and 3, cameras were set up to view behaviour within the brooding compartments, on external fixtures, and on the litter. For S3, cameras were mounted to view within the system (Figure 2c,d).

#### 2.3.2. Behaviour Observations

Noldus Observer XT version 14 (Noldus Information Technology, Wageningen, The Netherlands) was used for all behaviour observations. A total of 30 min of each recording was analysed using continuous focal observation [25]. Each recording was split into six observation periods of five minutes with the first beginning ten minutes after the recording had started. Once an observer completed a five-minute observation period, they would skip ahead eight minutes and begin the next observation period. Due to uncontrollable circumstances such as pullets knocking over cameras, some intervals between observation periods were decreased. Within each 5-min observation period, ten focal birds were followed for 30 s each [26] for a total of 60 focal birds observed per video.

Before beginning observations, sample videos from all aviary styles, locations, and visits were briefly scanned to assess spatial distribution and use of structures (described in Table 1). From these observations, a list of 10 pre-determined locations for focal bird selection was created to represent the approximate use of the system (e.g., middlemost pullet on the lowest perch, pullet 1/3 down from the left of the highest terrace, pullet 1/3 down from the right of the litter). To accommodate for differences in available fixtures, different lists were created for before and after the systems were opened, for the different aviary styles, as well as for different camera heights.

The closest pullet to the given focal location on the same fixture was selected. If no pullet was located on that fixture, a bird on the same fixture-type was selected (i.e., if the pre-determined location was “pullet in the middle of the highest perch”, but no bird was present, then the observer would select the middlemost pullet on the lowest perch). The behaviours of the focal bird were recorded based on the ethogram in Table 2. All occurrences of group walking and running were recorded, even if it did not include the focal bird. If a focal bird went out of view during the 30-s interval, the observer would select a new focal bird at the same start location.

Camera views used were based on visit and aviary style. At V1, when the brooding compartments were closed, cameras in locations B and E were used. This resulted in a maximum of three hours of footage. At V2 and V3, cameras with a high and low view of locations B and E, as well as footage from the wall and middle locations were used, for a maximum of six hours of footage analysed. Due to differences in housing design, in S3, cameras in locations B, wall, and aisle were used for a total of 1.5 hours of footage at V1 and 2 hours of footage at V2 and V3. If footage was missing, footage from an equivalent alternative camera was used.

Due to differences in the nature of the aviary styles (see Figure 1 and Figure 2), fewer videos were taken from S3 than S2 and S1. There was no equivalent middle litter section, and therefore no cameras D, E, or F. This amounted to a maximum of six fewer hours of video analysed per farm for S3 than S1 and S2.

Three observers analysed the videos. Observers could not be blind to the aviary styles or colour of the bird, but aside from main experimenter were blind to the hypotheses. Videos were watched in a random order according to a list created by an online list generator (https://www.random.org/lists/, accessed on 10 April 2019). Observers had to pass an interobserver reliability test with a Kappa of at least 0.60 before starting video analysis. The kappa coefficients for interobserver reliability ranged from 0.82 to 0.93. Intra-observer reliability was tested by having observers re-analyse the same video used for interobserver reliability after all video analysis was complete. The Kappa coefficients for intra-observer reliability ranged from 0.75 to 0.89.

### 2.4. Data Collection: Musculoskeletal Characteristics

#### 2.4.1. Sample Collection

Pullets were euthanised via cervical dislocation by trained staff at each farm during V2 and V3 (68.5 ± 4.94 and 111.5± 3.45 days of age, respectively). A total of 16 birds were euthanised from each flock, eight per visit. We were unable to obtain birds from several S1 flocks, limiting the S1 sample to one complete brown-feathered flock (V2 and V3), one complete white-feathered flock (V2 and V3), and eight pullets from V2 from a second white-feathered flock. This resulted in samples sizes of *n* = 40 for S1, *n* = 80 for S2 and *n* = 80 for S3. In S1 and S2, four birds were taken from a wall aisle, and four birds were taken from a middle aisle. In S3, birds were randomly selected throughout the aviary. Carcasses were stored at −20 °C. Before dissection, carcasses were thawed in a refrigerated room for approximately 48 hours.

#### 2.4.2. Bone and Muscle Collection

Bone and muscle collection were performed according to the methods of Casey-Trott et al. [9] by six trained individuals. The left pectoralis major, pectoralis minor, bicep, and the combination of all leg muscles (of the femur: iliotibialis, sartorius, semitendinosus, semimembranosus, quadriceps femoris, ambiens, adductor longus; of the tibiotarsus: gastrocnemius, tibialis anterior, peroneus longus, flexor perforans et perforates II and II). All muscles were weighed immediately after removal. The keel, left humerus, radius, femur, and tibia (with fibula attached) were also removed.

#### 2.4.3. Three-Point Bone Breaking Strength

After removal from the pullets, bones were stored in plastic bags at −20 °C until needed. Bones were thawed in a cooler for 24 hours before breaking. An Instron system (Model Material Testing, Norwood, MA) with Bluehill Universal software was used to measure bone breaking strength (BBS). Each bone was positioned in the same orientation on a cradle support with posts 4 cm apart for V2 and 5 cm apart for V3. A 5 kN load cell at a speed of 100 mm/min was used to apply force from the three-point bending test fixture (10 mm anvils and 50 mm in length) to the mid-point of the bone shaft. The maximum force required to break the bone was recorded (N).

#### 2.4.4. Keel Area

To account for different keel shapes, the total keel area and cartilage area were calculated using ImageJ 1.46r program (US National Institutes of Health Berthesda, Maryland, USA). Images of the keels were taken immediately after removal with a Canon Powershot Elph 340 Hs camera at a distance of 23.8 cm [20]. Images were uploaded to ImageJ and traced on an XP Pen Artist Pro 15.6in drawing tablet. The scale was re-set for each image between the same two points (average scale of 13.4 ± 0.90 pixels). The cartilage area was defined as the white area at the tip of the keel.

### 2.5. Statistical Analysis

All data were analysed using SAS 9.4 (SAS Institute Inc., Cary, NC) with a generalised linear mixed model (Proc GLIMMIX). For all analyses, flock was considered the experimental unit, with visit included as a repeated measure, and flock as a random effect. The statistical level of significance was set to *p* < 0.05.

For each response variable, the normality of residual variance was analysed with Proc Univariate. The model was assumed to be normally distributed if the Shapiro–Wilk statistic was above 0.80. A Gaussian distribution was initially used, and if it did not fit, a lognormal distribution was used.

#### 2.5.1. Behaviour Observations

For all behaviours, the model included visit, aviary style, feather colour, time of day, and all of their interactions as fixed effects. Time of day was not included as a repeated measure because Proc GLIMMIX could only include one repeated measure. Time of day and all of its interactions were included as fixed effects. Time of day was significant for the percentage of time standing and sitting. However, because time of day was only included in the model to account for its variation, the results are not presented here.

Durations of state behaviours were calculated as a percentage of the analysed time. New bird, area out of view, and no bird in location were not included in analysed time. Aerial behaviours were the combined sum of aerial ascent, aerial descent, and aerial across. Percentage of total time locomoting was calculated as the sum of walk, run, wing-assisted run (WAR), and all aerial behaviours. Percent total time stationary was the sum of stand and sit. Locomotor events were calculated as a rate per 30 min. All jumps and aerial transitions were combined into vertical transitions. All occurrences of group locomotion were recorded, even if the focal bird was not involved.

A Gaussian distribution was used for the percentage of time locomoting, stationary, standing, sitting, walking, as well as the rate of wing flaps, vertical transitions, aerial transitions, and jumps. A Lognormal distribution was used for the percentage of time running, WAR, performing aerial transitions, as well as the rate of group locomotion. Prior to the analysis, if the percentage of time or rate of locomotor events was zero, it was changed to 0.0001. The back-transformed values for all variables analysed with a lognormal distribution are shown except for the rate of group locomotion. For group locomotion exclusively, the lognormal distribution was used for hypothesis testing, but the LS Means from the Gaussian distribution are given in the tables.

#### 2.5.2. Musculoskeletal Characteristics

The model for all measures taken included visit, style, strain colour, and all of their interactions were included as fixed effects.

Individual muscle weights, keel area, and BBS were all corrected for pullet bodyweight by dividing the value by the pullet’s weight in kilograms.

A Gaussian distribution was used for all measures.

## 3. Results

### 3.1. Locomotion

#### 3.1.1. Main Effects

As interactions drove the main effects, only the results from the main effects, not part of a significant interactive effect, are presented below.

**Visit.** The percentage of total time locomoting (the sum of time spent walking, running, WAR, and performing aerial behaviours) differed between visits (*p* = 0.0007) (Table 3). Despite the brooding compartments being closed at V1, there was no difference in the percentage of time locomoting between V1 and V3. However, at V2, after the brooding compartments were opened, pullets spent significantly more time locomoting compared to V3. Visit number was the only significant factor affecting the percentage of time spent stationary (*p* = 0.0025). At V2, pullets spent the least amount of time stationary; however, this did not differ from the percentage of time spent stationary at V1 (Table 4).

There was a main effect of visit on the percentage of time pullets spent walking (*p* = 0.0068) and performing aerial transitions (*p* = 0.0007; Table 3). The rate of wing flaps (*p* = 0.0001), jumps (*p* = 0.0056), aerial transitions (*p* = 0.0007), and vertical transitions (*p* = 0.0003) were also significantly affected by visit (Table 5). Although pullets spent more time walking at V2 after the brooding compartments were opened, this was not different from V1. The percentage of time walking did, however, decrease between V2 to V3.

The percentage of time performing aerial transitions (Table 3) and the rate of wing flaps, and aerial and vertical transitions decreased throughout rearing, with significantly more locomotion at V1 and V2 compared to V3 (Table 5). Similarly, the rate of jumps decreased significantly between V1 and V3, with V2 being intermediate.

**Aviary Style.** The percentage of time locomoting was affected by aviary style (*p* = 0.0009; Table 3). Pullets in S3 spent a greater percentage of the observed time locomoting than pullets in S1 and S2. Despite these differences, time spent locomoting was still a small percentage of their total time budget, ranging from approximately 3.6% to 5.5% of observed time.

There was a main effect of aviary style on the percentages of time pullets spent walking (*p* = 0.0068) and performing aerial transitions (*p* = 0.0003; Table 3). In general, activity increased with increasing amounts of horizontal space, vertical space, and environmental complexity. The percentage of time spent walking was greater in S3 compared to S1 and S2. The percentage of time spent performing aerial behaviours was greater in S2 and S3 compared to S1.

**Colour.** There was a trend for strain colour to affect time locomoting (*p* = 0.0535), with white-feathered strains spending more time locomoting than brown-feathered strains (Table 3). Overall, strain colour alone affected two behaviours: percentage of time spent on aerial behaviours (*p* = 0.0321; Table 3) and the rate of jumps (*p* < 0.0001; Table 5). For both behaviours, white-feathered strains showed higher rates of activity compared to brown-feathered strains.

#### 3.1.2. Interactions

**Interaction between aviary style and visit number.** There was a significant interaction between visit and aviary style on the percentage of time spent standing (*p* < 0.0001), sitting (*p* < 0.0001), running (*p* = 0.0114), WAR (*p* = 0.0110), and the rate of group locomotion (*p* < 0.0001; Table 5). Compared to V1, pullets in all styles of aviaries spent more time standing during V3; during V1, pullets in S1 spent less time standing than all other combinations (Figure 3a)**.** At V1, pullets in S1 spent more time sitting than all other combinations, and at V3, pullets in all styles spent less time sitting than all other combinations except for flocks at V2 in S1 (Figure 3b). Over time, standing and sitting had an inverse relationship as shown in Table 4. The percentage of time spent standing increased, while the percentage of time spent sitting decreased. As mentioned earlier, pullets spent greater than 92% of their time stationary (sum of time standing and sitting).

The majority of active running and WAR for individual focal animals was observed during V1 in the S3 aviaries (Figure 3c,d). Bouts of group locomotion were most common at V1 in S3 than the other two styles or visits (Figure 3e); however, after the brooding compartments were opened (V2), the rate of total group locomotion in S1 and S2 increased such that there was no difference between styles. During V3, the rate of group locomotion decreased overall and was not different between styles.

**Interaction between visit and strain colour.** There was a significant interactive effect of visit and strain colour on the percentage of time spent standing (*p* = 0.0020), sitting (*p* = 0.0002), and WAR (*p* = 0.0007; Table 3 and Table 4). The percentage of time pullets spent sitting decreased across each visit regardless of colour (Figure 4a). The only exception was white-feathered strains during V3, which spent the same amount of time sitting as all strains at V2. Conversely, the percentage of time spent standing increased across each visit regardless of colour (Figure 4b).

At V1, white-feathered strains spent more time performing WAR than brown-feathered strains (Figure 4c). By V2, the time spent WAR decreased in white-feathered strains such that there was no difference between strain colour. The time spent WAR at V3 decreased for both strain colours and was neither different between strains nor from performing WAR at V2.

**Interaction between aviary style and strain colour.** The rate of wing flaps (*p* = 0.0012), aerial transitions (*p* = 0.0129), and vertical transitions (*p* = 0.0201) were affected by the interaction between aviary style and strain colour (Table 5). The rate of wing flaps (Figure 5a), aerial transitions (Figure 5b), and vertical transitions (Figure 5c) did not vary between strains within styles, except for S3, where white-feathered strains had a higher rate of all three locomotory events than all other combinations. However, the rate of wing flaps in S3 of white-feathered pullets did not differ from brown-feathered strains in S2. While lower in S1, the rate of aerial transitions did not differ between S2 and S1. White-feathered strains in S2 had a higher rate of aerial transitions than brown-feathered strains in S1.

### 3.2. Musculoskeletal Characteristics

The least-square means for the long bones’ BBS, muscles weights, and keel characteristics are shown in Table 6, Table 7 and Table 8, respectively. Measures are expressed as the proportion of body weight. As interactions drove main effects, only the results from main effects that are not part of an interactive effect are presented below.

#### 3.2.1. Main Effects

**Visit.** The BBS of the humerus (*p* < 0.0001) and femur were affected by visit (*p* < 0.0001; Table 6). Both bones were proportionally stronger at V2 compared to V3. Visit did not affect the proportional breaking strength of the radius (*p* = 0.3493). All muscle weights were significantly affected by visit number: pectoralis major (*p* < 0.0001), pectoralis minor (*p* < 0.0001), and leg muscle group (*p* < 0.0001; Table 7), with all muscles being proportionally heavier at V3 compared to V2. The percentage of keel cartilage was affected by visit number, with pullets at V2 having significantly more cartilage than at V3 (*p* < 0.0001; Table 8). Proportional keel area, however, was not affected by visit (*p* = 0.3005).

**Aviary Style.** BBS of the tibia (*p* = 0.0162) and femur (*p* = 0.0004) were affected by aviary style. The tibia was significantly stronger in S3 compared to S1, and the femur was significantly stronger in pullets housed in S3 compared to S1 and S2 (Table 6). Aviary style did not affect the breaking strength of the radius (*p* = 0.3888) or humerus (*p* = 0.1836). Aviary style did not affect the proportional weight of the pectoralis major (*p* = 0.9428), pectoralis minor (*p* = 0.4942), bicep (*p* = 0.1545), or leg muscle group (*p* = 0.8665).

**Strain Colour.** Strain colour affected the BBS of the femur (*p* < 0.0001), which was proportionally stronger in white-feathered strains than brown-feathered strains (Table 6). However, strain colour did not affect the breaking strength of the radius (*p* = 0.8898) or humerus (*p* = 0.1836). Strain colour did affect proportional weights of the pectoralis major (*p* < 0.0001) and leg muscle group (*p* < 0.0001), but not the bicep (*p* = 0.4269). While the pectoralis major was proportionally heavier in white-feathered strains than brown-feathered strains, the leg muscle group was proportionally lighter in white-feathered strains compared to brown-feathered strains (Table 7). Keel area and cartilage percent were both significantly affected by strain colour. The white-feathered strain had a proportionally larger keel area (*p* = 0.0003) and less cartilage (*p* < 0.0001) than brown-feathered strains (Table 8).

#### 3.2.2. Interactions

There was a visit by colour interaction that affected proportional tibia BBS (*p* < 0.0001; Table 6) and bicep weight (*p* = 0.0153; Table 7). While there was no difference between strain colours at V2, the tibia BBS of both white- and brown-feathered strains at V2 was significantly stronger than those at V3 (Figure 6a). At V3, white-feathered strains had proportionally stronger tibiae than brown-feathered strains. Furthermore, at V3, brown-feathered strains had proportionally heavier biceps than both white- and brown-feathered strains at V2 (Figure 6b).

Finally, there was a significant interactive effect of aviary style and strain colour on the pectoralis minor weight (*p* = 0.0104; Table 7). Within strain colour, there were no differences between styles; however, white-feathered strains in S2 had significantly heavier pectoralis minor muscles than brown-feathered strains regardless of aviary style. White-feathered strains in S1 and S3 also had significantly heavier pectoralis minor muscles than brown-feathered strains in S2 and S3 (Figure 7).

## 4. Discussion

### 4.1. Aviary Style Effects on Locomotion and Musculoskeletal Characteristics

Rearing aviary style affected locomotion and the BBS of the tibia and femur. Considering that pullets in S3 spent the most time locomoting, and in particular white-feathered strains performed the highest rate of aerial and vertical transitions, it follows that their leg bones would be stronger. Judex and Zernicke [18] found similar results in that the tarsometatarsus of immature roosters who performed more jumps had increased bone formation compared to the controls. Furthermore, pullets in S3 spent more time walking than those in S1 or S2. In humans, walking is the most common bone-loading activity as it provides load from both ground-reaction forces and strain from muscle contraction against the bones [27]. As bone formation is site-specific to where the load is applied [15,16], the observed differences in locomotion were likely sufficient to increase bone formation of the leg bones, which increased their strength.

Interestingly, there was no difference in the BBS of the radius or humerus, muscle weights, or keel characteristics between aviary styles. This is particularly unexpected as pullets in S3 performed the most wing-associated behaviours (WAR, aerial transitions, and wing-flaps). A previous study showed that wing use involves the pectoral muscles, which provides mechanical load to the keel [19]. To date, individual behaviours have not been tested as to their effects on the keel and pectoral muscle development. However, Casey-Trott et al. [9] hypothesised that the opportunity for increased wing-associated behaviours contributed to the greater lateral keel area of aviary-reared pullets compared to conventionally reared pullets. Furthermore, aviary-reared pullets had a greater percentage of cartilage at 16 weeks of age than conventionally-reared pullets, which may have been an indication of a longer period of growth and slower ossification in aviary-reared pullets. Those authors found that aviary-reared pullets also had stronger radii and humeri, as well as proportionally heavier pectoral and bicep muscles than conventionally-reared pullets. However, the magnitude of the difference in opportunity to perform wing-related locomotion between the conventional rearing cage and rearing aviary in the Casey-Trott et al. study [9] was considerably greater than that between the different rearing aviary styles in our study, especially after the brooding compartments were opened. Conventional rearing cages offer little space for pullets to even flap their wings, let alone use them when running or during aerial transitions. Although we did see differences in wing-related locomotory behavior between pullets in the different styles of rearing aviaries, these behavioural differences were likely not of the magnitude to result in musculoskeletal differences in the keel, wings or pectoralis muscles.

As expected of a trial conducted at multiple commercial facilities, there were many different management practices involved. Light intensity varied greatly between flocks, but since it was not measured, it is not possible to determine if one style had an overall higher or lower intensity than another. Although light intensity may contribute to different activity levels, a recent study comparing white- and brown-feathered strains reared at 10, 30, and 50 lux found no difference in overall active behaviours (standing, walking, jumping, and flying) throughout rearing [28]. Therefore, it is unlikely that the variation in light intensity had a large effect on the present study’s results.

There were confounds of some management factors between aviary styles. Natural light was provided for two S2 flocks, one S3 flock, but no S1 flocks. Furthermore, natural light was only provided in the morning, during the first recording at each visit. Organic production was also confounded with style as there were no S1 organic flocks, one S2 and two S3 organic flocks. Considering that there was no difference in the percentage of time locomoting between S1 and S2, it is possible that the provision of natural light and organic production did not significantly affect locomotion of these two rearing aviary styles. Finally, due to logistical reasons, visits were not done at the exact same ages across all farms. However, there was no statistical difference in the average age of each visit across styles. Despite all of these differences in management, there were still significant differences in locomotion and musculoskeletal characteristics between styles.

### 4.2. Strain Feather Colour Effects on Locomotion and Musculoskeletal Characteristics

The BBS of the tibia and femur was greater in white-feathered strains compared to brown-feathered strains. However, the tibiae of white-feathered strains were only stronger than those of brown-feathered strains at V3. It is possible that the increased overall rate of jumps and vertical transitions for white-feathered strains throughout rearing increased bone formation of the tibia and femur as a result of the ground reaction forces from landing [18]. Alternatively, correcting BBS values for pullet body weight may have created these differences as brown-feathered strains are heavier than white-feathered strains at 16 weeks of age [29,30]. In future studies, it may be beneficial to compare different methods of correction for BBS, such as correcting for the bone area or bone weight to see if these findings remain consistent.

A recent study by Fawcett et al. [20] found similar results as the present study. They compared the proportional BBS and muscle weights of different strains of pullets reared in enriched cages (the same system used as S2, but with the doors remaining closed throughout rearing). They reported that sixteen-week-old Dekalb White and Lohmann Select Leghorn-Lite pullets had proportionally stronger tibiae and femurs than Lohmann Brown pullets. Again, agreeing with the present findings, the authors reported that the white-feathered pullets had proportionally larger keels and pectoral muscles, as well as lighter leg muscles than the brown-feathered strains. Jensen [21] evaluated the behaviour of the same pullets at 14 weeks of age and found no difference in the locomotion between strain feather colours; however, white-feathered strains were found to use perches more than brown-feathered strains. The white-feathered strains in the present study performed more wing-associated behaviours, thus utilised their pectoral muscles more than the brown-feathered strains [19]. The differences in proportional pectoral muscle weight and keel area between strain feather colours were, therefore, likely due to genetic differences in body conformation and exaggerated by the differences in locomotion.

### 4.3. Age Effects on Locomotion and Musculoskeletal Charactersitics

As would be expected, the proportional muscle weights and BBS of the long bones differed between visits. All of the muscles examined became proportionally heavier between V2 and V3. The only exception was the bicep of white-feathered strains, which did not differ in proportional weight between the two visits. Interestingly, despite the proportional increase in pectoral muscle weight between visits, there was no difference in the proportional keel area. This discrepancy may be in part due to correcting the values with the pullets’ bodyweight. Another possibility is that the locomotion performed by the pullets increased muscle growth but was insufficient to cause equally proportional bone formation of the keel.

Between V2 and V3, the proportional BBS of the humerus, radius, and femur all decreased. Similar to the keel, it is possible that the types of locomotion performed were sufficient to increase proportional muscle growth but did not cause equally proportional bone loading over time. There was also a decrease in locomotion between V2 and V3, which may be another explanation for the decreased BBS. The reduction in locomotion would have decreased bone-loading both from mechanical loading and muscle contractions, thus not stimulating as much bone formation over time relative to body growth [31,32]. A more likely explanation, however, is the growth trajectories for different body components. Typically the skeletal frame begins to develop before significant muscle deposition occurs, and it is when the pullets are approximately eight weeks of age that muscle deposition begins to increase [29,30].

As mentioned, the percentage of time locomoting significantly decreased between V2 and V3, with the percentage of time pullets spent locomoting peaking at V2. The increased time pullets spent locomoting between V1 and V2 may be due to the increase in free space [33] created after the brooding compartments were opened. While there was no significant visit by style interaction for overall time locomoting, the open-concept of S3 provided pullets with the opportunity to perform more high-intensity locomotion than pullets in S1 and S2 whose brooding compartments were closed. Specifically, chicks in S3 at V1 spent more time running and WAR, as well as performed a higher rate of group locomotion, a behaviour which pullets could not perform in S1 and S3 at that time. However, during V2 and V3, after the brooding compartments were opened, there was no difference in running, WAR, or group locomotion between aviary styles.

There was a significant decrease in locomotion between V2 and V3, which is consistent with previous findings. Kozak et al. [34] reported a decrease in locomotion between 10- and 16-week-old pullets. The observed decrease in locomotion between V2 and V3 was likely a result of maturation [34] but may have also been affected by the effective stocking density, as the birds grew and increased in body size filling the available free space [21,35]. Increasing stocking density of pullets in an enriched pullet cage (the same system as S2, but with the compartments remaining closed throughout rearing) was found to decrease the time 14-week-old pullets spent locomoting. In an adult perchery system, increased crowding has been shown to decrease flock locomotion and increase time individual hens spent standing [35].

Similarly, in the present study, as the pullets grew and became more crowded, there was a significant increase in time spent standing. Riddle et al. [36] found that 28-week-old Dekalb white hens need approximately 3445 cm^2^ to wing flap and 567 cm^2^ to lie down. However, the Canadian Codes of Practice state that pullets reared in multi-tiered systems such as the ones used in this study require a minimum of only 342 cm^2^ of floor space [1]. Brown-feathered strains also take up significantly more horizontal space than white-feathered strains [37], which may result in them being more affected by crowding. It should be noted, however, that the oldest pullets in this study were around 16 weeks of age; therefore, they would require less space to wing flap and lie down than 28-week-old hens. Furthermore, pullets in the present study may have been active (e.g., eating, drinking, preening, perching) while stationary, so levels of activity may have been underestimated. Age (maturity) and degree of crowding were also confounded due to the nature of the study, so it cannot be concluded which of these two factors may have contributed the most to the decrease in locomotion between V2 and V3.

## 5. Conclusions

Despite variation in management factors across commercial flocks, there were significant differences found between the age of pullets, strain colours, and aviary styles. Overall, younger pullets, white-feathered strains, and pullets housed in the most complex aviary style (S3) locomoted the most. The rate of vertical transitions was the most notable difference as white-feathered strains and pullets in S3 performed the highest rate. The increased proportional BBS of the leg bones of those pullets compared to brown-feathered strains and those in the least complex aviary style (S1) may be in part due to their increased rate of vertical transitions. As there were no obvious associations in individual locomotory behaviours, the heavier pectoral muscles, lighter leg muscles, and larger keel area of white-feathered compared to brown-feathered strains were likely due to differences in body conformation between genetic strains.

These results suggest that white-feathered strains and pullets reared in most complex S3 rearing aviary had improved locomotory and musculoskeletal development compared to brown-feathered strains and pullets reared in the less complex S1 and S2 aviaries. Therefore, strain as well as differences in rearing aviary design can affect the types of physical activities that growing pullets perform, which may, in turn, impact their skeletal development.

## Figures and Tables

**Figure 1 animals-11-00634-f001:**
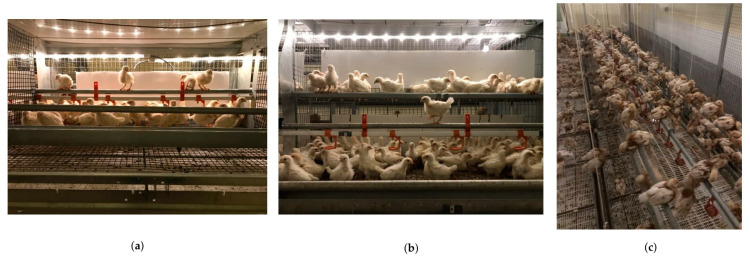
Brooding compartment for (**a**) style one; (**b**) style two; (**c**) and style three rearing aviaries.

**Figure 2 animals-11-00634-f002:**
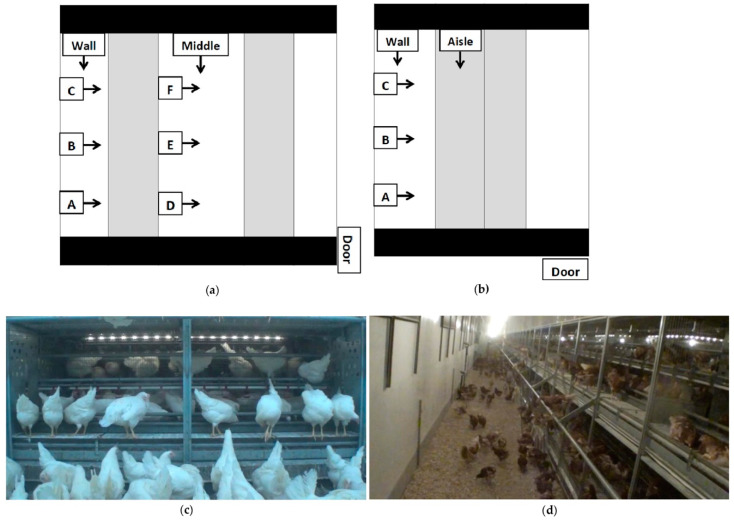
Aerial view of camera placements for **(a)** Style 1 and Style 2; **(b)** Style 3 rearing aviaries; **(c)** still shot from a low view into a Style 2 brooding compartment; **(d)** a still shot from a “wall” long view of a Style 2 rearing aviary. Blacked out areas are walkways for human traffic, unoccupied by pullets. The shaded sections are the aviary structures, and the white section is the litter area. Black lines separate rows of aviary structures. Cameras (represented by boxes) were located at the brooding compartment or panel section ¼ (A,D), ½ (B,E) and ¾ (C,F) down the aisle. At each letter location, one camera was positioned to view the litter floor and the bottom tier, and one viewed the top tiers of the system. Camera-labelled wall and middle viewed the litter area lengthwise. In style three, the camera-labelled aisle viewed the system lengthwise. Arrows indicate the direction in which cameras were facing.

**Figure 3 animals-11-00634-f003:**
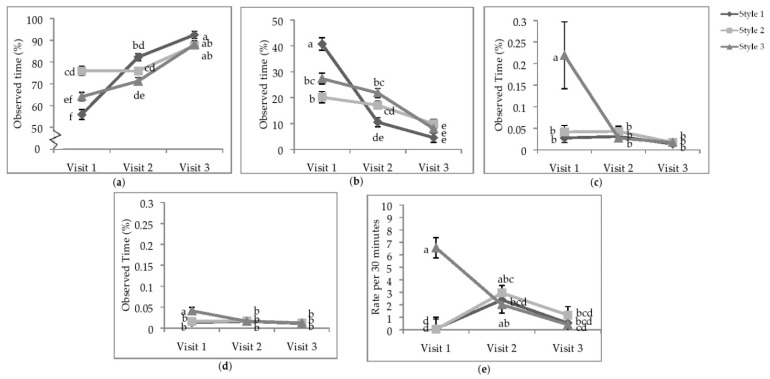
Least squares means of the visit by style interaction of state behaviours. Percentage of time (**a**) standing; (**b**) sitting; (**c**) running; (**d**) wing-assisted running; (**e**) the rate per 30 min of group locomotion. Visits one, two, and three were when the pullets were 25.9 ± 6.67; 68.0 ± 4.78 days of age; 112.1 ± 3.34 days of age. During visit one, the brooding compartments were closed, whereas during visits two and three, the brooding compartments were open into the litter areas. Error bars represent standard error. ^a–f^ Indicate significant differences between means (*p* < 0.05).

**Figure 4 animals-11-00634-f004:**
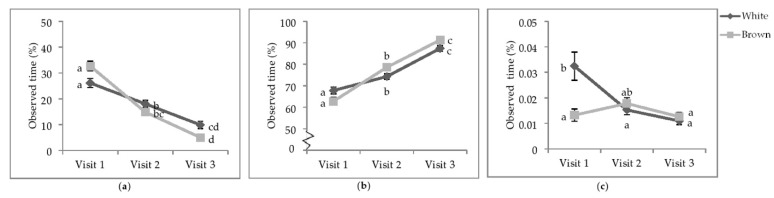
Least square means of the visit by colour interactions of state behaviours. Percentage of observed of time (**a**) sitting; (**b**) standing; (**c**) wing-assisted running. Visits one, two, and three were when the pullets were 25.9 ± 6.67; 68.0 ± 4.78 days of age; 112.1 ± 3.34 days of age. During visit one, the brooding compartments were closed, whereas during visits two and three, the brooding compartments were open into the litter areas. Error bars represent standard error. ^a–c^ Indicate significant differences between means (*p* < 0.05).

**Figure 5 animals-11-00634-f005:**
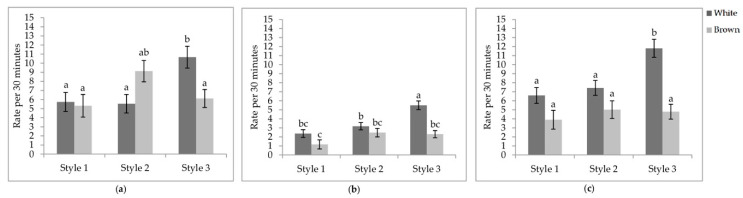
Least square means of the style by colour interactions of locomotory events. Rate per 30 min of (**a**) wing flaps; (**b**) aerial transitions; (**c**) vertical transitions. Visits one, two, and three were when the pullets were 25.9 ± 6.67; 68.0 ± 4.78 days of age; 112.1 ± 3.34 days of age. During visit one, the brooding compartments were closed, whereas during visits two and three, the brooding compartments were open into the litter areas. Error bars represent standard error. ^a–c^ Indicate significant differences between means (*p* < 0.05).

**Figure 6 animals-11-00634-f006:**
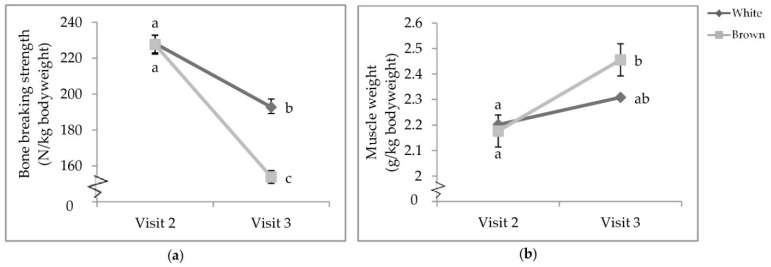
Visit by colour interaction of (**a**) tibia breaking strength; (**b**) bicep weight. Pullets were 68.5 ± 4.94 and 111.5 ± 3.45 days of age, respectively, when euthanised by cervical dislocation during visits two and three. Error bars represent standard error.

**Figure 7 animals-11-00634-f007:**
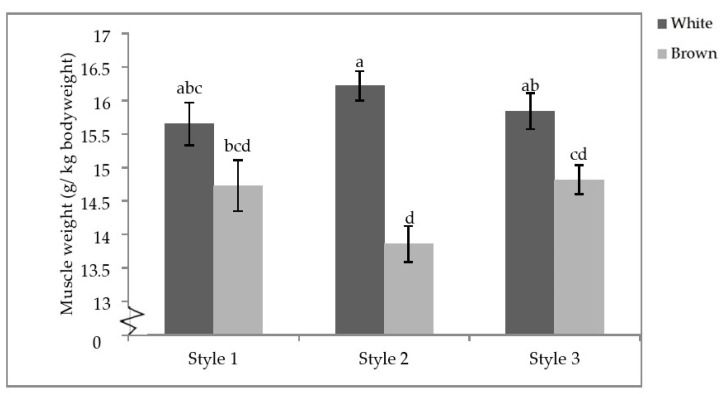
Aviary style by strain colour interaction of pectoral minor weights. Error bars represent standard error. ^a–d^ Indicate significant differences between means (*p* < 0.05).

**Table 1 animals-11-00634-t001:** Description of different locations available to pullets in the rearing aviaries.

Location	Description
Litter	Floor outside of the system where substrate has been placed. Only available after the brooding compartments have been opened.
System floor	Wire or plastic grid floor directly above a manure belt.
Platform	Raised wire or slatted floors above a manure belt.
Terrace	Raised wire or slatted floors not above a manure belt, usually above the litter.
Perch	A round or square bar, either inside or outside the system, onto which birds grip with their feet.
Ramp	An angled slatted or wire structure placed at an angle beside the system, typically joining the litter to a terrace or upper tier.

**Table 2 animals-11-00634-t002:** Ethogram used for observations.

Behaviour	Definition	Method of Recording
Stand	Pullet has both legs extended straight underneath her body; her head is in an upright position. Pullet may be performing other behaviours such as pecking, eating, drinking, or preening.	Duration
Sit	Pullet’s abdomen contacts with the floor with both legs under the body and head out. Pullet may be performing other behaviours such as pecking or preening.	Duration
Walk	Pullet raises one of her legs with the other leg on the floor and moves forward for at least three steps. Head and neck bob as the pullet moves.	Duration
Run	Pullet raises one of her legs with the other leg on the floor and moves forward rapidly for at least three steps. Head is outstretched with no head or neck movement.	Duration
Wing-assisted run	Pullet runs while moving her wings upwards and downwards. The forward movement may involve a short jump.	Duration
Aerial transition	Pullet moves with both legs simultaneously off the floor, either increasing, decreasing, or remaining on the same plane of elevation. Wings are fully extended to assist in the transition.	Duration and count
Jump	Pullet moves with both legs simultaneously off the floor, either increasing, decreasing, or remaining on the same plane of elevation. Wings are not used to assist in the transition. Typically, a very quick transition.	Count
Wing flap	Pullet extends her wings fully and moves them up and down rapidly.	Count
Group locomotion	A group of ten or more pullets walk or run to one end of the system (Style 3 only) or litter (all Styles). A new bout begins when a group of ten or more pullets run in the opposite direction. All occurrences recorded.	Count
Other	Pullet performs a behaviour not included in the ethogram, or that cannot be categorised under a listed behaviour (e.g., play fighting and dust bathing).	Duration
No bird in location	No bird is present on any of the given focal location fixtures.	Duration excluded
Location out of view	View of focal location is obstructed typically due to dim lighting or other pullets blocking the location.	Duration excluded
New Bird	Short period of time to select a new focal bird.	Duration excluded

**Table 3 animals-11-00634-t003:** Least square means (±standard error) of the percentage of total time brown- and white-feathered strains of layer pullets housed in three different styles of commercial rearing aviary systems of increasing structural complexity spent on locomotory behaviours throughout the rearing period.

Effect	Behaviour, % Total Time (±SE)									
	Locomoting ^1^		Walking		Running		Wing-Assisted Running		Aerial Transitions	
	Mean	*p*-Value	Mean	*p-*Value	Mean	*p-*Value	Mean	*p*-Value	Mean	*p*-Value
**Visit Number ^2^**										
l	**4.7 (0.58) ^ab^**		**4.3 (0.53) ^ab^**		0.06 (0.014) ^a^		0.02 (0.003) ^a^		**0.13 (0.025) ^a^**	
2	**5.4 (0.50) ^a^**	0.0065	**5.2 (0.46) ^a^**	0.0068	0.03 (0.006) ^a^	0.0004	0.02 (0.002) ^a^	0.002	**0.10 (0.016) ^a^**	0.0003
3	**2.7 (0.51) ^b^**		**2.6 (0.47) ^b^**		0.02 (0.003) ^b^		0.01 (0.001) ^b^		**0.04 (0.007) ^b^**	
**Aviary Style**										
S1	**3.6 (0.41) ^a^**		**3.5 (0.38) ^a^**		0.02 (0.004) ^a^		0.01 (0.002) ^a^		**0.05 (0.008) ^a^**	
S2	**3.8 (0.39) ^a^**	0.0009	**3.5 (0.36) ^a^**	0.0016	0.03 (0.005) ^ab^	0.0053	0.02 (0.002) ^ab^	0.0365	**0.10 (0.014) ^b^**	<0.0001
S3	**5.5 (0.39) ^b^**		**5.1 (0.36) ^b^**		0.05 (0.007) ^b^		0.02 (0.002) ^b^		**0.13 (0.021) ^b^**	
**Strain Colour**										
White	4.7 (0.32)	0.0535	3.6 (0.31)	0.0633	0.03 (0.006)	0.8092	0.02 (0.002)	0.1297	**0.10 (0.013) ^a^**	0.0321
Brown	3.8 (0.33)		4.4 (0.30)		0.03 (0.004)		0.01 (0.001)		**0.07 (0.009) ^b^**	
**Interactions** ^3^										
Visit × Style		0.1462		0.2562		**0.0114**		**0.0110**		0.1450
Visit × Colour		0.6134		0.5994		0.9167		**0.0007**		0.8698
Style × Colour		0.0820		0.0784		0.8346		0.1216		0.0895

Bolded means indicate significantly different main effects not driven by interactions. ^a–b^ Means with different letters within a column indicate a significant effect of aviary style, visit, or strain colour on percent total time spent performing each behaviour (*p* < 0.05). ^1^ Sum of time spent walking, running, wing-assisted running, and performing aerial transitions. ^2^ Visit 1 was at 25.9 ± 6.67 days of age; visit 2 was at 68.0 ± 4.78 days of age; visit 3 was at 112.1 ± 3.34 days of age. ^3^ Significant interactions drove the individual fixed effects.

**Table 4 animals-11-00634-t004:** Least square means (±standard error) of the percentage of total time brown- and white-feathered strains of layer pullets housed in three different styles of commercial rearing aviary systems of increasing structural complexity spent on stationary behaviours throughout the rearing period.

Effect	Behaviour, % Total Time (±SE)					
	Stationary ^1^		Standing		Sitting	
	Mean	*p*-Value	Mean	*p-*Value	Mean	*p-*Value
**Visit Number** ^2^						
l	**94.8 (0.77) ^ab^**		65.3 (1.20) ^a^		29.4 (1.28) ^a^	
2	**92.9 (0.66) ^a^**	0.0025	76.5 (0.92) ^b^	<0.0001	16.5 (0.98) ^b^	<0.0001
3	**96.8 (0.69) ^b^**		89.4 (0.96) ^c^		7.4 (1.01) ^c^	
**Aviary Style**						
S1	95.5 (0.68)		76.9 (1.22) ^ab^		18.6 (1.40)	
S2	95.5 (0.66)	0.0575	79.8 (1.48) ^a^	0.0049	15.7 (1.33)	0.1577
S3	93.5 (0.66)		74.4 (1.16) ^b^		19.1 (1.34)	
**Strain Colour**						
White	94.5 (0.53)	0.3849	76.5 (0.94)	0.4275	18.0 (1.08)	0.7513
Brown	95.2 (0.56)		77.6 (0.99)		17.5 (1.14)	
**Interactions** ^3^						
Visit × Style		0.2046		**<0.0001**		**<0.0001**
Visit × Colour		0.4370		**0.0020**		**0.0002**
Style × Colour		0.2414		0.8196		0.6833

Bolded means indicate significantly different main effects not driven by interactions. ^a–c^ Means with different letters within a column indicate a significant effect of aviary style, visit, or strain colour on percent total time spent performing each behaviour (*p* < 0.05). ^1^ Sum of time spent standing and sitting. ^2^ Visit 1 was at 25.9 ± 6.67 days of age; visit 2 was at 68.0 ± 4.78 days of age; visit 3 was at 112.1 ± 3.34 days of age. ^3^ Significant interactions drove the individual fixed effects.

**Table 5 animals-11-00634-t005:** Least square means (±standard error) of the rate per 30 min of wing-associated behaviours performed by brown- and white-feathered strains of layer pullets housed in three different styles of commercial rearing aviary systems of increasing structural complexity throughout the rearing period.

	Behaviour, Rate Per 30 min (±SE)									
Effect	Wing Flap		Aerial Transition		Jump		Vertical Transition ^1^		Group Locomotion	
	Mean	*p*-Value	Mean	*p-*Value	Mean	*p-*Value	Mean	*p*-Value	Mean	*p-*Value
**Visit Number ^2^**										
l	**9.6 (0.84) ^a^**		**3.5 (0.37) ^a^**		**4.6 (0.45) ^a^**		**8.1 (0.67) ^a^**		1.5 (0.30)	
2	**7.5 (0.70) ^a^**	0.0001	**3.3 (0.30) ^a^**	0.0007	**3.7 (0.32) ^ab^**	0.0056	**7.0 (0.52) ^a^**	0.0003	0.76 (0.243)	0.0838
3	**4.1 (0.73) ^b^**		**1.6 (0.31) ^b^**		**2.9 (0.33) ^b^**		**4.6 (0.54) ^b^**		0.18 (0.252)	
**Aviary Style**										
S1	5.5 (081) ^a^		1.8 (0.33) ^a^		3.5 (0.48)		5.2 (0.68) ^a^		0.19 (0.253) ^a^	
S2	7.3 (0.78) ^ab^	0.0377	2.8 (0.31) ^a^	<0.0001	3.4 (0.45)	0.2257	6.2 (0.64) ^ab^	0.0040	0.60 (0.236) ^a^	0.0002
S3	8.4 (0.78) ^b^		3.9 (0.31) ^b^		4.4 (0.46)		8.3 (0.65) ^b^		1.7 (0.24) ^b^	
**Strain Colour**										
White	7.3 (0.59)	0.6145	3.7 (0.25) ^a^	<0.0001	**4.9 (0.37) ^a^**	<0.0001	8.6 (0.52) ^a^	<0.0001	0.51 (0.202)	0.1987
Brown	6.8 (0.66)		2.0 (0.27) ^b^		**2.6 (0.39) ^b^**		4.6 (0.55) ^b^		1.1 (0.19)	
**Interactions** ^3^										
Visit × Style		0.1715		0.2717		0.2838		0.2841		**<0.0001**
Visit × Colour		0.0698		0.5617		0.4437		0.4422		0.4186
Style × Colour		**0.0012**		**0.0129**		0.1309		**0.0201**		0.0746

Bolded means indicate significantly different main effects not driven by interactions. ^a–b^ Means with different letters within a column, indicate a significant effect of aviary style, visit, or strain colour on percent total time spent performing each behaviour (*p* < 0.05). ^1^ Combined rate of jumps and aerial transitions. ^2^ Visit 1 was at 25.9 ± 6.67 days of age; visit 2 was at 68.0 ± 4.78 days of age; visit 3 was at 112.1 ± 3.34 days of age. ^3^ Significant interactions drove the individual fixed effects.

**Table 6 animals-11-00634-t006:** Least square means (±standard error) of the maximum bone breaking strength (N/kg) of the radius, humerus, tibia, and femur of brown- and white-feathered strains of layer pullets housed in three different styles of commercial rearing aviary systems of increasing structural complexity at approximately 10 and 16 weeks of age.

	Maximum Bone Breaking Strength, N/kg ^1^ (±SE)							
Effect	Radius		Humerus		Tibia		Femur	
	Mean	*p*-Value	Mean	*p-*Value	Mean	*p-*Value	Mean	*p*-Value
**Visit Number** ^2^								
2	32.3 (0.49)	0.3493	**182.9 (2.33) ^a^**	<0.0001	227.9 (3.42) ^a^	<0.0001	**232.7 (3.73) ^a^**	<0.0001
3	31.8 (0.52)		**141.3 (2.55) ^b^**		172.2 (2.83) ^b^		**154.0 (4.01) ^b^**	
**Aviary Style**								
S1	32.6 (0.90)		166.6 (3.60)		**191.4 (4.49) ^a^**		**185.8 (5.62) ^a^**	
S2	32.3 (0.65)	0.3888	158.5 (2.53)	0.1836	**196.8 (3.18) ^ab^**	0.0162	**187.3 (3.89) ^a^**	0.0004
S3	31.2 (0.65)		161.2 (2.54)		**206.5 (3.33) ^b^**		**207.0 (3.86) ^b^**	
**Strain Colour**								
White	32.0 (0.57)	0.8898	164.5 (2.34)	0.1571	209.7 (3.13) ^a^	<0.0001	**211.0 (3.58) ^a^**	<0.0001
Brown	31.1 (0.64)		159.7 (2.40)		187.1 (2.98) ^b^		**175.7 (3.82) ^b^**	
**Interactions ^3^**								
Visit × Style		0.7189		0.1155		0.5096		0.0695
Visit × Colour		0.5338		0.6468		**<0.0001**		0.1577
Style × Colour		0.5951		0.4548		0.8439		0.0623

Bolded means indicate significantly different main effects not driven by interactions. ^a–b^ Means with different letters within a column indicate a significant effect of aviary style, visit, or strain colour on percent total time spent performing each behaviour (*p* < 0.05). ^1^All maximum bone breaking strength values were adjusted for pullet bodyweight. ^2^ Pullets were euthanised at 68.5 ± 4.94 at visit 2, and 111.5 ± 3.45 days of age at visit 3. ^3^ Significant interactions drove the individual fixed effects.

**Table 7 animals-11-00634-t007:** Least square means (±standard error) of the proportional muscle weight (g/Kg) of the pectoralis major, pectoralis minor, bicep, and leg muscle group of layer pullets housed in three different styles of commercial rearing aviary systems of increasing structural complexity at approximately 10 and 16 weeks of age.

	Muscle Weights, g/kg ^1^ (±SE)							
Effect	Pectoralis Major		Pectoralis Minor		Bicep		Leg Muscle Group ^2^	
	Mean	*p*-Value	Mean	*p-*Value	Mean	*p-*Value	Mean	*p*-Value
**Visit Number** ^3^								
2	**41.6 (0.51) ^a^**	<0.0001	**13.6 (0.15) ^a^**	<0.0001	2.2 (0.04) ^a^	0.0001	**74.9 (0.75) ^a^**	<0.0001
3	**49.2 (0.55) ^b^**		**16.7 (0.16) ^b^**		2.4 (0.04) ^b^		**83.4 (0.80) ^b^**	
**Aviary Style**								
S1	45.4 (0.76)		15.2 (0.25)		2.4 (0.08)		79.3 (1.15)	
S2	45.5 (0.52)	0.9428	15.0 (0.17)	0.4942	2.3 (0.06)	0.2798	78.8 (0.80)	0.8665
S3	45.3 (0.52)		15.3 (0.17)		2.2 (0.06)		79.3 (0.79)	
**Strain Colour**								
White	**47.9 (0.48) ^a^**	<0.0001	15.9 (0.16) ^a^	<0.0001	2.3 (0.05)	0.4269	**77.0 (0.73) ^a^**	<0.0001
Brown	**42.9 (0.51) ^b^**		14.5 (0.17) ^b^		2.3 (0.06)		**81.3 (0.79) ^b^**	
**Interactions** ^4^								
Visit × Style		0.2244		0.5859		0.1628		0.4446
Visit × Colour		0.6566		0.4264		**0.0153**		0.8855
Style × Colour		0.0939		**0.0104**		0.1545		0.9312

Bolded means indicate significantly different main effects not driven by interactions. ^a–b^ Means with different letters within a column indicate a significant effect of aviary style, visit, or strain colour on percent total time spent performing each behaviour (*p* < 0.05). ^1^ All muscle weight values were adjusted for pullet body weight. ^2^ Leg muscle group comprised of all femur and tibiotarsus muscles (of the femur: iliotibialis, sartorius, semitendinosus, semimembranosus, quadriceps femoris, ambiens, adductor longus; of the tibiotarsus: gastrocnemius, tibialis anterior, peroneus longus, flexor perforans et perforates II and III). ^3^ Pullets were euthanised at 68.5 ± 4.94 at visit 2, and 111.5 ± 3.45 days of age at visit 3. ^4^ Significant interactions drove the individual fixed effects.

**Table 8 animals-11-00634-t008:** Least square means (±standard error) of the proportional keel area (mm^2^/kg) and percentage of keel cartilage of brown- and white-feathered strains of layer pullets housed in three different styles of commercial rearing aviary systems of increasing structural complexity at approximately 10 and 16 weeks of age.

Effect	Keel Area, mm^2^/kg ^1^ (±SE)		Cartilage, % ^2^ (±SE)	
	Mean	*p*-Value	Mean	*p-*Value
**Visit Number** ^3^				
2	1639.0 (29.10)	0.3005	**29.0 (0.47) ^a^**	< 0.0001
3	1677.0 (31.09)		**15.6 (0.50) ^b^**	
**Aviary Style**				
S1	1704.9 (51.73)		22.6 (0.89)	
S2	1665.4 (36.62)	0.2355	21.3 (0.64)	0.1573
S3	1603.5 (36.58)		23.0 (0.64)	
**Strain Colour**				
White	**1748.0 (32.74) ^a^**	0.0003	**19.1 (0.56) ^a^**	< 0.0001
Brown	**1567.8 (36.17) ^b^**		**25.4 (0.63) ^b^**	
**Interactions** ^4^				
Visit × Style		0.7696		0.2128
Visit × Colour		0.2825		0.7596
Style × Colour		0.4749		0.1307

Bolded means indicate significantly different main effects not driven by interactions. ^a–b^ Means with different letters within a column indicate a significant effect of aviary style, visit, or strain colour on percent total time spent performing each behaviour (*p* < 0.05). ^1^ Keel area was corrected for pullet bodyweight. ^2^ Percent cartilage determined by drawn area. ^3^ Pullets were euthanised at 68.5 ± 4.94 at visit 2, and 111.5 ± 3.45 days of age at visit 3. ^4^ Significant interactions drove the individual fixed effects.

## Data Availability

Data is contained within the article and Appendix A.

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
