# Peer review of "Effects of Rearing Aviary Style and Genetic Strain on the Locomotion and Musculoskeletal Characteristics of Layer Pullets"

_animals, 2021, doi:10.3390/ani11030634_

Round 1

Reviewer 1 Report

Very well written paper. It is very easy to follow and understand. I thoroughly enjoyed reading it.

I have a minor comment:

  1. Line 470: As explained from a study in line 463-464, its the use of particular parts such as wing that provides mechanical loading and strength to the bones. And, in this study birds in S3 performed more wing related behavior. The increased wing related behavior has also resulted in larger keel and pectoral muscles (line 501). So, from the results in this study, does that mean that just the opportunity to use their wings are enough to develop the strength? The duration or number of times does not have significance? Do you think there was a difference in bone formation due to difference in activity, however, not enough to show the difference in strength?

Reviewer 2 Report

The paper evaluates a field trial looking at the locomotion and musculoskeletal characteristics of aviary pullets in three different types of rearing systems. The content is appropriate and well executed. There are very minor comments that would help the presentation of the paper.

The supplemental material is appreciated but would like to see in section 2.2 a range of stocking densities for the flocks evaluated. 

Figure 1. The different colored lines are hard to distinguish when printed in color and would doubt anyone could make them out if printed in black and white. I would suggest finding another way to denote the differences among the pictures.

Figure 2. The letters and words in (a,b) can not be discerned when printed. Increase the font size.

Tables 1,2 - revisit justification of text (left not centered) to make the tables easier to read.

Ln 295. WAR? It is not defined prior to using this abbreviation.

Tables 3-5, need to provide more detail in the title. If I pull the tables out of the paper, there is no context to what animals, ages, anything is being reported.

Figure 4 - space needed before Error

Tables6,7 - New titles as the title doesn't reflect the content of the tables.

Overall - very well written paper and no criticism on approach, methodology, statistics or interpretation of the data. 

Reviewer 3 Report

Review: “Effects of Rearing Aviary Style and Genetic Strain on the Locomotion and Musculoskeletal Characteristics of Layer Pullets”

General comments:

I enjoyed reading your manuscript. It is well written and has a good structure. The topic, the effect of aviary style during rearing, is interesting. Previous research has focused mainly on the difference between cage and aviary rearing and not so much on different aviary styles. It would be very interesting if your study followed the flocks during egg production to investigate the same parameters, end of lay.

In general, I find the M&M section to be very long. In addition, I lack information regarding age of the birds. Belove you will find my specific comments.

Specific comments:

Line 31: please include how locomotion was evaluated and scored.

Line 54-55: you list KBF as the first welfare challenge. Secondly, you list navigation and dehydration. However, most of your investigation and recordings are related to locomotion and musculoskeletal factors other than the keel and navigation. I suggest that you restructure the first paragraph of your introduction, to make it fit your main aim and the issues you actually investigated.

In general, your “Materials and methods”-section is 6.5 pages. This is a bit much. Can you make it shorter?

Line 111: “according to individual farm management”. Did this vary much between farms? I do believe that lighting regime may affect the results.

Line 112: “near barn capacity”. What does mean in terms of stocking density?

Line 198: in what time period was the 30 minutes recording? Morning, midday, evening? Was it the same time for all flocks?

Line 196: you did a lot of behavioural observations. This is not reflected in your abstract nor in your aim.

Line 285: you have named this heading “Locomotion”. Would it not be more correct to call it “Behavioural observations” or something like that?

Line 318: I assume that time locomoting is assessed in the same time period during the day? The time budget for hens may vary significantly during the day.

Line 318: the term “Visit” is a bit confusing to me. Are the different visits linked to the age of the birds? So V1 is the youngest birds and V3 the oldest birds? If so, is it not the age of the bird and not the visit that creates the difference? It is more logical and easier to explain biologically if you refer to the age of the birds. Was visiting-age the same for all flocks? Please include information about the bird age during each visit

Line 468: what is a “larger” keel? Longer? Broader? Does this affect the ossification?

In Line 476:  you state that light intensity was not measured. Do you think that different intensities may have affected your results and locomotion score? This should be discussed in more detail.

Line 509: “Age Effects”. This is related to my comment above: I can not see that you any where state the age of the birds. This is important information that should be included. Is all birds in the same age during visit 1,2 and 3?

Line 539: please state stocking density at the last visit

Round 2

Reviewer 3 Report

Thank you for the respons to my comments. I have no further remarks.